# Effect of OH-Group Introduction on Gas and Liquid Separation Properties of Polydecylmethylsiloxane

**DOI:** 10.3390/polym15030723

**Published:** 2023-01-31

**Authors:** Evgenia A. Grushevenko, Tatiana N. Rokhmanka, Ilya L. Borisov, Alexey V. Volkov, Stepan D. Bazhenov

**Affiliations:** 1A.V. Topchiev Institute of Petrochemical Synthesis, Russian Academy of Sciences, Leninsky Prospect 29, 119991 Moscow, Russia; 2Biological and Environmental Science, and Engineering Division (BESE), Advanced Membranes and Porous Materials Center (AMPM), King Abdullah University of Science and Technology, Thuwal 23955, Saudi Arabia

**Keywords:** polydecylmethylsiloxane, polymer design, pervaporation, gas separation, olefine-aldehyde separation, carbon dioxide removal

## Abstract

Membrane development for specific separation tasks is a current and important topic. In this work, the influence of OH-groups introduced in polydecylmethylsiloxane (PDecMS) was shown on the separation of CO_2_ from air and aldehydes from hydroformylation reaction media. OH-groups were introduced to PDecMS during hydrosilylation reaction by adding 1-decene with undecenol-1 to polymethylhydrosiloxane, and further cross-linking. Flat sheet composite membranes were developed based on these polymers. For obtained membranes, transport and separation properties were studied for individual gases (CO_2_, N_2_, O_2_) and liquids (1-hexene, 1-heptene, 1-octene, 1-nonene, heptanal and decanal). Sorption measurements were carried out for an explanation of difference in transport properties. The general trend was a decrease in membrane permeability with the introduction of OH groups. The presence of OH groups in the siloxane led to a significant increase in the selectivity of permeability with respect to acidic components. For example, on comparing PDecMS and OH-PDecMS (~7% OH-groups to decyl), it was shown that selectivity heptanal/1-hexene increased eight times.

## 1. Introduction

Polymer design is an important and popular direction in membrane science. Such an approach allows a polymeric membrane to be obtained with predefined separation properties. In general, the design of membrane materials is reduced to functionalization with the intent of controlling the relative rate of diffusive or solubilitygases in the membrane [1]. The choice of the main polymer chain is largely determined by the scientific novelty of the work, as well as by the stability of that polymer in the separating medium. The approach of introducing functional groups into membrane-forming polymers is widespread among researchers [2,3,4,5]. The introduction of oxygen-containing groups increases the membrane affinity to acidic components and its hydrophilicity. In particular, it is known that the introduction of hydroxyl functional groups increases the proton conductivity of fuel cell membranes [6]. The introduction of ether groups can increase the selectivity of carbon dioxide extraction from the gas mixture. In [7], an increase in CO_2_/CH_4_ selectivity was demonstrated by introducing cellosolve moieties into the polynorbornenes, moreover, a high selectivity (31) of n-butane/methane pair separation was maintained due to the “alkyl tail” length. The introduction of ether-functionalized pyridinium-based ionic liquids into the cellulose acetate-based membrane achieved a seven-fold increase in CO_2_ permeability with CO_2_/N_2_ and CO_2_/CH_4_ permselectivities of 32 and 24, respectively [8].

Membrane gas separation technology is rapidly gaining popularity and is of great interest to industry as a promising way to separate gas mixtures [9]. For decarbonization tasks, membrane processes are sufficiently competitive compared with traditional separation methods for CO_2_ emissions, especially for its capture after the combustion of biogas, as well as for carbon dioxide removal from natural gas [10]. The most commonly used membrane polymers for this type of problem are polysulfone [11,12], polyamide [13,14] and polydimethylsiloxane (PDMS) [15]. Due to their high chain mobility, PDMS-based membranes are characterized by high permeability, which makes them attractive from a practical point of view [16,17]. In this regard, the development of composite membranes based on PDMS is relevant, since the mechanical properties of self-supported membranes based on it are insufficient. In particular, PDMS is used for the finish coating of gas separation membranes in order to improve their gas separation properties by eliminating the defects of the selective layer [18]. For example, in [19], the coating of PDMS on a polysulfone membrane increased the CO_2_/N_2_ selectivity from 4.5 to 4.7. When CO_2_ was removed from a mixture with nitrogen or hydrogen using tubular membranes with a PDMS selective layer in [20], CO_2_/N_2_ and CO_2_/H_2_ selectivity values of 8.6 and 3.8, respectively, were obtained, and CO_2_ permeability reached 3.08·10^−7^ cm^3^cm^−2^s^−1^Pa^−1^.

Chemical modification of polysiloxanes can improve their CO_2_-fility. Researchers in [21] showed that the introduction of a bulky polar side group CF_3_CH_2_CH_2_- into the main chain of siloxane increased the CO_2_/CH_4_ selectivity to a value of 5.6. Introduction of amino groups in PDMS allowed an increase in selectivity by carbon dioxide of up to two times, with a comparable permeability through the membrane [22].

Membranes based on polysiloxanes are of great scientific and practical interest, in particular for hydrocarbon capture and hydrophobic pervaporation [4], due to their high permeability and thermal stability [23]. In addition, they are chemically stable in a large number of solvents due to their cross-linked structure [24,25]. The separation ability of polydimethylsiloxane (PDMS) and polyoctylmethylsiloxane (POMS) membranes has been extensively studied [26,27,28,29]. Chemical modification of the polysiloxane chain can improve the separation characteristics of membranes. In particular, the hydrophobization of polysiloxanes is of great interest to researchers. Thus, the introduction of alkyl side substituents of lengths 7, 8 and 10 into the polysiloxane chain allowed an increase in the separation factor of 1% wt methyl-tretbutyl ether in water from 111 (for the methyl group) to 161, 169 and 180, respectively [30]. The introduction of 10% diacetate groups into the polysiloxane chain allowed an increase in the phenol flow from aqueous solution by 15% compared with polydimethylsiloxane, which led to an increase in the phenol/water separation factor to 21.5 [31]. When separating a mixture of methylisobutyl ketone—water, it was shown that the introduction of long alkyl substituents (octyl, tridecyl) led to a significant increase in separation efficiency: the separation factor increased from 705 for PDMS to 1030 (octyl) and 1200 (tridecyl) [31]. For problems of organophilic pervaporation, hydrophilization of the polysiloxane chain can be one of the approaches to increase the separation efficiency of hydrophobic and hydrophilic mixtures. For example, dispersion of 1-dodecanol in a PDMS matrix allowed the separation factor of the ethanol/water mixture to increase by 20% [32]. One promising direction in the development of membrane materials based on polysiloxanes is their use as a selective layer in a catalytic membrane for separation of hydroformylation reaction products [4,33]. Two principal configurations of the membrane hydroformylation reactor are currently proposed: the nanofiltration reactor [34,35] and the tubular reactor [36,37]. Nanofiltration of organic media has long been proposed as an alternative method to extraction and distillation to separate reaction products from the homogeneous hydroformylation catalyst [38]. For example, [39] describes the use of commercial nanofiltration membranes Starmem 122 and Starmem 240 of Grace Davison Company in the hydroformylation of 1-octene and dodecene, respectively. The membranes showed a decrease in permeability by 15 and 50% over time, which the authors explained was due to blocking of the membrane pores. The use of tubular membrane hydroformylation reactors will intensify this process by combining the stage of chemical reaction and release of its products [40]. Various scientific groups are actively developing hydroformylation catalysts immobilized on liquid or solid carriers [41,42]. The use of a membrane reactor in which a porous membrane substrate serves as a catalyst carrier and a selective layer provides noncontinuous removal of aldehydes from the reaction zone will increase the overall performance of the process. Thus, in [40], tubular membranes based on silicon carbide with an outer selective layer of PDMS were used for continuous gas phase hydroformylation of 1-butene. At 100 °C and a feed gas pressure of 11 bar, this membrane showed a retention of 0.23 and permeability of 0.05 for 1-butene, and a retention of 0.61 and permeability of 0.29 for pentanal [40]. Thus, we can expect that the transition from PDMS to side-chain-substituted polymethylsiloxane will increase the selectivity of aldehyde extraction, including from the mixture with olefins, which is of great scientific and practical interest.

In this work, for the first time, the influence of the OH-group in the side-chain of the polydecylmethylsiloxane (PDecMS) was investigated. Taking into account the high perspective for polysiloxanes in gas and liquid separation and the common effect of the OH-group in membrane materials, CO_2_ removal from nitrogen was chosen as the gas separation task, and separation of the aldehyde/olefine mixture (reaction mixture of hydroformylation processes) as the liquid separation task.

## 2. Materials and Methods

### 2.1. Materials

The reagents used for polymer synthesis were polymethylhydrosiloxane (PMHS) (Mn = 1900 g/mol, ABCR, Karlsruhe, Germany); 1-decene (95% wt, Sigma-Aldrich, St. Louis, MO, USA); isooctane (chemical grade, Component Reactiv, Moscow, Russia, RF); 1,7-octadiene (95%, Sigma-Aldrich, St. Louis, MO, USA); 1,3-divinyl-1,1,3,3-tetramethyl disiloxane platinum complex (0), a solution in xylene (Sigma-Aldrich, St. Louis, MO, USA); polydimethylsiloxane vinyl terminated (PDMS) (Mn = 25,000 g/mol, Sigma-Aldrich, USA); 10-undecenol-1 (99%, ABCR, Karlsruhe, Germany); and chloroform (chemical grade, Chemmed, Podolsk, Russia, RF).

For membrane characterization, the following were used: 1-hexene (95% wt, Sigma-Aldrich, St. Louis, MO, USA); 1-heptene (95% wt, Sigma-Aldrich, St. Louis, MO, USA); 1-octene (95% wt, Sigma-Aldrich, St. Louis, MO, USA); 1-nonene (Tokyo Chemical Industry CO, Tolyo, Japan); ethanol (96% wt, A grade, Component Reagent, Mocsow, RF); as well as individual gases O_2_ (99.5% vol., MGPZ, Moscow, Russia), CO_2_ (99.5% vol., MGPZ, Moscow, Russia), N_2_ (99.5% vol., MGPZ, Moscow, Russia), and CH_4_ (99.5% vol., MGPZ, Moscow, Russia).

Some properties of the studied liquids are presented in Table 1.

### 2.2. Membrane Materials Synthesis

PDecMS was obtained by the following technique. The PMHS was mixed with a 15 wt % solution of 1-decene in isooctane and 10 wt % solution of 1,7-octadiene in isooctane, and in the presence of Carstead catalyst (1,3-divinyl-1,1,3,3-tetramethyl disiloxane platinum (0) complex), the solution in xylene was stirred for 2 h at 60 °C. Then PDMS with Mn = 25,000 g/mol in isooctane was added to the solution. Stirring of the reaction mixture was continued for one hour. After that, 3 wt % PMHS solution in isooctane was added to the reaction mixture to the stoichiometric ratio.

For obtaining OH-modified PDecMS, the described procedure was changed. PMHS was mixed with 15 wt % 1-decene solution in isooctane and with 13 wt % 10-undecenol-1 solution in chloroform (10-undecenol-1:1-decene = 10:90) and stirred for 2 h at 60 °C in the presence of Carstead catalyst. Then PDMS with Mn = 25,000 g/mol in isooctane in the molar ratio (1-decene + 10-undecenol-1):PDMS = 5 was added to the solution. Stirring of the reaction mixture was continued for one hour. After that, a 3 wt % solution of PMHS in isooctane was added to the reaction mixture to the stoichiometric ratio. Polymer films of OH-modified PDecMS were obtained by pouring the polymer solution on the surface of glycerol and subsequently drying in a heating oven (FED 115, Binder, Tuttlingen, Germany). A schematic illustration is shown in Figure 1.

### 2.3. Development Flat Sheet Composite Membrane

Flat sheet composite membranes based on PDecMS and OH-PDecMS were obtained by deep coating on fluoroplastic microfiltration porous support MFFK-1 (Vladipor, Vla-dimir, Russia). For coating the porous substrate, the viscosity of the polysiloxane polymer solution was brought to 35 mPa·s. The application was performed on a stage with a tape-retracting mechanism that provided one-sided contact of the tape of the porous substrate with the polymer solution [43]. To prevent polymer flow into the pores of the substrate, it was impregnated with ethanol and water in series. After application, the tape was moved to a drying table, where the cross-linking of polysiloxane to form a composite membrane was performed under conditions of natural convection at 60 °C. The constant mass of the membrane was reached after 4 h of drying, indicating the formation of a selective layer of the composite membrane. The schematic representation of the coating technic is shown in Figure 2.

### 2.4. Differential-Scanning Calorimetry (DSC)

Calorimetric measurements were performed using a differential-scanning calorimeter DSC 3+ (Mettler Toledo, Greifensee, Switzerland) at a speed of change of 10 deg/min in an argon atmosphere, in a temperature interval from −140 to 100 °C.

### 2.5. Nuclear Magnetic Resonance (NMR)

High-resolution 1H NMR spectra were obtained for solutions in CDCl_3_ according to the standard procedure, on a Bruker AVANCE III HD 400 NMR spectrometer (Karlsruhe, Germany).

### 2.6. Scanning Electron Microscopy (SEM)

SEM was used to characterize the structure and morphology of the membranes. SEM was carried out on a Thermo Fisher Phenom XL G2 Desktop SEM (Thermo Scientific, Waltham, MA, USA). Cross-sections of the membranes were obtained by fracturing in liquid nitrogen after preliminary impregnation of the specimens in isopropanol. A thin (5–10 nm) gold layer was deposited on the prepared samples in a vacuum chamber (~0.01 mbar) using a desktop magnetron sputter “Cressington 108 auto Sputter Coater” (Cressington Scientific Instruments Ltd., Watford, U.K.). The accelerating voltage during image acquisition was 15 kV.

### 2.7. Gas Transport Properties Measurments

Gas permeabilities and diffusion coefficients of membrane materials for individual gases (N_2_, CO_2_, CH_4_) were measured at 30 °C according to Daynes–Barrer technique using the precise unit “Helmholtz-Zentrum Geesthacht” mounted with a pressure sensor (“Baratron”), with an accuracy of 10^−7^ bar. Permeability was given in Barrer. Pure gas permeability coefficient *P_i_* was determined by the variable pressure/constant volume method, the diffusivity *D_i_* was measured according to the time-lag method, and the solubility coefficient *S_i_* was estimated as $S_{i}=P_{i}/D_{i}$. The ideal selectivity for gas pair n-butane and methane is given by the following relationship:(1)$$\alpha =P_{C_{4}H_{10}}/P_{{\mathrm{CH}}_{4}}=\left(D_{C_{4}H_{10}}/D_{{\mathrm{CH}}_{4}}\right)\cdot \left(S_{C_{4}H_{10}}/S_{{\mathrm{CH}}_{4}}\right)=\alpha_{D}\cdot \alpha_{S}$$
where, $\alpha_{D}$ is diffusivity selectivity, and $\alpha_{S}$ is solubility selectivity. The experimental error of the measurements of *P* was 5%, and of *D* was 9%.

Gas permeation of the composite membranes was determined by variable volume/constant pressure method. All experiments were conducted at room temperature (23.6–25 °C) and upstream pressure 0.5–4.5 atm. Downstream membrane pressure was atmospheric (near 1 atm). The flat sheet membrane was placed in stainless steel module and the active membrane area was 18.1 cm^2^. Permeating flux was measured by bubble mass flowmeter in the lab set-up (Figure 3). The permeability of the composite membrane was determined from the first derivative of the dependence of the permeating flow on the pressure drop. The standard deviation of experimental data was below 5%.

### 2.8. Sorption Measurments

Determination of the sorption of olefins and aldehydes in PDecMS and OH-PDecMS was performed from the mixture with solvent. The technique is described in detail in [44] and shortly represented below. Polymer film with fixed mass *m*_0_ was exposed in solvent mixture for 48 h in isothermal condition. The sorption of aldehydes and olefins in polysiloxanes in mixture with solvent was studied from binary solutions with ethanol. In the hydroformylation reaction, the catalyst was dissolved in a liquid carrier. Ethanol is used in some hydroformylation systems as a solvent for the catalyst [45,46,47]. The choice of ethanol in this work was due to two factors: the components under study (C_7_–C_10_ aldehydes and C_6_–C_9_ olefins) were dissolvable in it, and its sorption interaction with PDecMS was small compared with the liquids under study. The validity of this approach was described in [30], in which water was taken as a solvent and its zero sorption in polysiloxanes was the main assumption. Each experimental determination of sorption in PDecMS was performed three times. Measurements were taken at temperatures of 30–60 °C. The binary solution and the polymer under study were thermostatted for 48 h. The sorbate concentration in the initial liquid and in the liquid after 48 h of sorption was measured to determine the value of sorption. The concentration of each solution was determined as the average between three analytical measurements. The measurements were performed on a gas chromatograph Crystallux-4000M (NPF Meta-chrom, RF) equipped with an FID detector. Separation was performed using a Phenomenex Zebron ZB-FFAP capillary column, USA (length 50 m, diameter 0.32 mm, phase thickness 0.50 µm) with the phase being a copolymer of 2-nitroterephthalic acid ester and ethylene glycol in the following temperature regime: column—120 °C, detector—150 °C, evaporator—200 °C. The solubility coefficient (*S_i_*, mole solvent/(m^3^ polymer·kPa)) was calculated using Formula (2), similar to the calculation scheme described in detail in [30]:(2)$$k_{i}=\frac{n_{s}}{V_{pol}\cdot p_{sorb}}$$
where, *p_sorb_*—pressure over solution determined in Aspen plus (NRTL) for a solution with a concentration of w_48_, *n_s_*—the amount of sorbate substance in the polymer, and *V_pol_*—polymer volume.

The amounts of ethanol sorbed into the polymer ($n_{EtOH}^{S}$, mol) in the initial solution ($n_{EtOH}^{0}$, mol) and in solution after 48 h of sorption ($n_{EtOH}^{48}$, mol), were calculated by Formulas (3), (4) and (5), respectively.
(3)$$n_{EtOH}^{S}=S_{EtOH}\cdot V_{pol}$$
where, *S_EtOH_*—sorption of ethanol in the polymer, determined by exposing the polymer sample to ethanol at a given temperature for 48 h.
(4)$$n_{EtOH}^{0}=\frac{m_{p-pa}\cdot \left(1-w_{0}\right)}{M_{EtOH}}$$
where, $w_{0}$—mass concentration of sorbate in the initial solution; $m_{p-pa}$—initial solution mass, g; and *M_EtOH_*—ethanol molecular mass, g/mol.
(5)$$n_{EtOH}^{48}=n_{EtOH}^{0}-n_{EtOH}^{S}$$

The deviation of the obtained values from the average did not exceed 15%.

### 2.9. Study of Aldehyde and Olefin Transport through PDecMS-Based Membranes in the Vacuum Pervaporation Mode

The transport properties of the PDecMS membrane for 1-hexene (at 30, 60 °C) and heptanal (at 30, 60 and 90 °C) were studied in the vacuum pervaporation mode. The separation of binary solutions of 1-hexene (25% wt) in ethanol, and heptanal (45% wt) in ethanol, were studied. The studies were carried out in a laboratory vacuum pervaporation unit described elsewhere [30,44]. The effective area of the membrane was 13.5 cm^2^.

Total permeate flux (J, kg/m^2^h) was calculated by Formula (6):(6)$$J=\frac{m}{S\cdot t}$$
where, *m*—total mass of the permeate (kg) penetrating through the membrane of area *S* (m^2^), for a known period of time *t* (h).

Separation factor (*α*) was determined by Formula (7):(7)$$\alpha =\frac{y_{o}\cdot x_{w}}{y_{w}\cdot x_{o}}$$
where, *x_o_* and *x_w_*—mass fraction of the organic component and ethanol in the mixture to be separated; and *y_o_* and *y_w_*—mass fraction of the organic component and ethanol in the permeate.

Partial flux of the component (*J_i_*, kg/m^2^h) was calculated by Formula (8):(8)$$J_{i}=\frac{m_{i}}{S\cdot t}$$
where, *m_i_*—mass of the mixture component in the permeate (kg) penetrating through the membrane of area *S* (m^2^), in the time interval *t* (h).

Permeability coefficient (*P*, mol·m/(m^2^·h·kPa)) for component *i* was calculated according to Formula (9):(9)$$P_{i}\;=\;\frac{J_{i}l}{\left(P_{i}^{f}-P_{i}^{p}\right)}$$
where, *l*—selective layer thickness, m; and $p_{i}^{f}$ and $p_{i}^{p}$—vapor pressure of component *i* in the initial mixture and in the permeate (kPa), respectively.

Membrane selectivity ($\alpha_{ij}{}^{m}$) was determined from the ratio:(10)$$\alpha_{ij}{}^{m}=\frac{P_{i}}{P_{j}}$$

To account for the contribution of evaporation to the pervaporation separation process, the phase transition separation factor was calculated (*α_v_*):

(11)$$\alpha_{v}=\frac{C_{wv}^{M}\cdot C_{wf}^{W}}{C_{wv}^{W}\cdot C_{wf}^{M}}$$
where, $C_{wv}^{M}$ and $C_{wv}^{W}$—mass fraction of penetrant and ethanol in saturated vapor above the solution to be separated, and $C_{wf}^{M}$ and $C_{wf}^{W}$—mass fraction of penetrant and ethanol in the solution to be separated.

To determine the vapor pressure of the permeate and the initial mixture, the activity coefficients were calculated using the model NRTL (non-random two-liquid) using the software package Aspen Plus 10.

Enthalpy of sorption $∆H_{A}^{s},\;\left(\mathrm{kJ}/\mathrm{mol}\right)$ was calculated from Formula (12), and the permeability activation energy $∆E_{A}^{p}$, (kJ/mol) was determined from Formula (13):(12)$$k=k_{0}\cdot e^{\frac{-∆H_{A}^{s}}{R\cdot T}}$$
where, *R*—molar gas constant, equal to 8.31 J/(mol·K); and *T*—temperature, K.
(13)$$P=P_{0}\cdot e^{\frac{-E_{A}^{p}}{R\cdot T}}$$

## 3. Results

### 3.1. Membrane Material Characterization

#### 3.1.1. Polymer Properties

Based on the ratio of reagents in the hydrosilylation reaction in the modified sample of OH-PDecMS, it should contain about 7% of OH-groups. This concentration of undecenol substituents in the polysiloxane allowed for the obtainment of a homogeneous polymer film. Increasing the 1-undecenol concentration in the reaction media led to the obtainment of heterogeneous material. To determine their actual content in the polymer solution prior to drying, NMR analysis was performed (Figure 4).

By ^1^H NMR spectroscopy of pre-cross-linked polymers, it was shown that a chemical shift of 7.3 ppm corresponded to CDCl_3_ (solvent). The difference in integral intensity for OH-PDecMS (Figure 4 (1)) and PDecMS (Figure 4 (2)) near 5 ppm shift which could be attributed to the terminal hydrogen atom in CH_2_=CH- group was caused by the variant cross-linking degree of the polysiloxane. Moreover, the deferens of integral intensity for Si-H (characteristic shift—4.77–4.79 ppm [48]) also corresponded to the variant cross-linking degree of polymers. For OH-PDecMS spectra, a peak with a chemical shift of 3.69 ppm designated to H atom from the -CH_2_- group of the aliphatic chain bonded with the -OH group. The peak integral was 0.0109. The presence of a chemical shift of 1.32 ppm (-(-CH_2_-)-), 0.58 ppm (-Si-CH_2_- [48]), and triplet at 0.94 ppm (-CH_3_ group in aliphatic chain) for both polymers showed that hydrosilylation between PHMS and 1-decene occurred. The peak integrals for OH-PDecMS and PDecMS were 0.5259 and 0.4636, respectively. Additionally, the characteristic shift at 0.11 ppm could be attributed to the Si-CH_3_ group [49]. This shift was observed for OH-PDecMS and PDecMS.

To identify the ratio of undecyl-1-ol to decyl in OH-PDecMS, the peak integrals were calculated for shifts 0.94 (-CH_3_ in the aliphatic chain) and 3.69 (-CH_2_- in the aliphatic chain bonded with the -OH group). Their values were 0.1123 and 0.0050, respectively. It should be taken into account that peak integral needed to be normalized regarding the H atom amount in the considered group (0.1123/3 and 0.0050/2). Therefore, the amount of OH-substituted side groups was 6.3% in OH-PDecMS. The value obtained correlated well with the mass balance of the synthesis (6.5 mol % undecyl-1-ol groups).

Thermograms of the synthesized polysiloxanes showed a characteristic appearance for cross-linked comb-like polysiloxanes (Figure 5): glass transition temperatures were in the negative region, the observed melting peak corresponded to the phase formed by side substituents, similar to [13]. The introduction of undecenol groups in PDecMS led to a change in the glass transition temperature of the polymer and the melting of the region formed by the side groups. Thus, for the PDecMS synthesized in this paper, T_g_ = −67.9 °C (close to the value for the PDecMS first presented in [14], T_g_ = −68.3 °C). In the case of the OH-PDecMS sample modified with undecenol groups, a decrease in the glass transition temperature to T_g_ = −69.4 °C was observed. The melting temperatures (T_m_) of the side substituents were practically identical (T_m_ = −38.5 °C (OH-PDecMS) and −38.9 °C (PDecMS)), which was not surprising since in OH-PDecMS the majority of the side groups were decyl (in PDecMS all side substituents were 1-decene). Therefore, we concluded that the obtained materials were very close in thermal properties.

#### 3.1.2. Gas Transport Properties of the Membrane Materials

To determine the mass transfer through the obtained membrane materials, the permeability and diffusion coefficients of individual gases were determined. Permeability (P) and diffusion (D) coefficients were investigated for pure CO_2_, N_2_ and CH_4_ in membrane materials PDecMS and OH-PDecMS (Table 2). Solubility coefficients were calculated as the ratio of the permeability coefficient to the diffusion coefficient. For comparison, the data for PDMS [50] were used.

The obtained membrane materials were characterized by high coefficients of gas permeability by carbon dioxide, while the coefficients of permeability of nitrogen and methane were 2–3 times lower. In general, this indicated the selectivity of these materials to CO_2_. By comparison with PDMS, the permeability coefficients of the materials obtained in this work were lower. The decrease in permeability coefficients in comparison with PDMS was associated with a decrease in diffusion coefficients: a decrease of 3–4 times, depending on the gas, was observed. Diffusion coefficients of CO_2_ for PDecMS and OH-PDecMS differed by 15%. The increase in CO_2_ diffusion with the introduction of a volume substituent with a hydroxyl group in PDecMS is associated with an increase in the free volume available for CO_2_ transfer. It is known that the diffusion diameter for CO_2_ (0.302 nm) is smaller than for N_2_ (0.304 nm) and CH_4_ (0.318 nm) [51]. In contrast, the calculated solubility coefficient values of the materials obtained in this work were higher than those of PDMS (except for nitrogen). The introduction of the side substituent in the polysiloxane led to an increase in the contribution of the sorption component to the mass transfer through the membrane. It should be noted that the introduction of OH-groups into PDecMS led to an almost two-fold increase in the solubility coefficient of CO_2_—from 2.60·10^−2^ to 4.74·10^−2^ cm^3^/(cm^3^·cm Hg). Based on the permeability, diffusion, and solubility coefficients obtained, the ideal permeability, diffusion, and solubility selectivities were calculated (Table 3).

OH-PDecMS demonstrated the highest permselectivity for both CO_2_/N_2_ and CO_2_/CH_4_, in comparison with PDecMS and PDMS. It should be known that the value of permselectivity is determined by diffusion selectivity. The same result was reported in [52] for n-butane/methane gas pair and explained by features of the packing of side groups of polydecylmethylsiloxane. In comparison with the PDMS membrane, materials based on PDecMS had a higher value of sorption selectivity for CO_2_/N_2_ gas pair, and a higher value of diffusion selectivity for CO_2_/CH_4_ gas pair.

In the hydroformylation reaction, for the membrane reactor for which the use of PDecMS and OH-PDecMS were proposed, aldehydes were produced by the conversion of olefins in a synthesis gas current. Diffusion diameters of CO_2_ and N_2_ were the same (0.304 nm), which allowed for the simulation of carbon monoxide transfer without carrying out direct measurements for safety reasons. In [33], an approach to estimating the permeability of synthesis gas through PDMS was shown, and it was noted that hydrogen penetrated the membrane 2–4 times faster than nitrogen. It is worth noting that synthesis gas transport through PDecMS and OH-PDecMS is determined by the diffusion component of transport, which performs a smaller contribution to the mass transfer in polysiloxanes, whereas the transport of aldehydes is determined by the sorption interaction with the polysiloxane, creating a decisive contribution to the permeability value. In view of the above, it could be assumed that the transfer of synthesis gas through the membrane would not be high, but its depletion in hydrogen could be observed.

#### 3.1.3. Sorption Hydrophobic and Hydrophilic Liquids in PDecMS and OH-PDecMS

In order to determine the influence of OH on interactions with liquids of different nature, the sorption of olefins (1-hexene, 1-heptene, 1-octene, 1-nonene) and aldehydes (heptanal, decanal) were studied, and solubility coefficients (S) were calculated at temperatures of 30–60 °C. The task of separation of these substances was the recovery of hydroformylation reaction products from the reaction zone [38]. The main difficulty of this separation task was that C_n+1_ aldehydes are formed from C_n_ olefins. Aldehydes are low-volatile high-boiling liquids, so their separation from the mixture with catalysts and higher-boiling olefins is not possible by direct distillation. As the molecular weight increases, the boiling point naturally increases, while the saturated vapor pressure decreases manifold. It was expected that such a low volatility of the compounds under study would render it difficult for them to diffuse through the membrane. Accordingly, the sorption interaction of olefins and aldehydes with the membrane material would determine their transmembrane mass transfer. The solubility coefficients of aldehydes and olefins in PDecMS and OH-PDecMS at the investigated temperatures are presented in Figure 6.

The general trend for OH-PDecMS and PDecMS was a decrease in the solubility coefficients of olefins and aldehydes with an increase in temperature from 30 to 60 °C. This phenomenon in polymer systems has been widely studied [53]. It is worth noting that solubility coefficients increase with the increasing molecular weight of the sorbate, including solubility coefficients of aldehydes in polysiloxanes higher than solubility coefficients of olefins. When comparing the solubility of these substances in OH-PDecMS and PDecMS, it should be noted that while the solubility coefficients of olefins were comparable, the solubility coefficients of aldehydes differed by an order of magnitude in the temperature range under study. For example, at 30 °C for PDecMS, values of S (1-hexene) = 0.016 ± 0.002 mol/(ml·bar) and S (heptanal) = 0.88 ± 0.13 mol/(ml·bar) were obtained; and for OH-PDecMS, values of S (1-hexene) = 0.017 ± 0.003 mol/(ml·bar) and S (heptanal) = 7.4 ± 1.1 mol/(ml·bar) were obtained. Such a strong sorption interaction of OH-PDecMS with aldehydes could be attributed to the presence of -OH groups. The group contribution of hydrogen bonding energy for the -OH bond is 20,000 J/mol, and for the -CH_3_ bond is 0 J/mol [54]. Since aldehydes, unlike olefins, are characterized by a significant contribution of hydrogen bond interaction [55], then the greater interaction parameter of these compounds would be with the OH-PDecMS polymer. Such a difference in the thermodynamic interaction of aldehydes with PDecMS and OH-PDecMS determined the difference in their solubility selectivity (Table 4).

It is worth noting that the solubility selectivities of the polymers under study differed by an order of magnitude (in proportion to the solubility coefficients of aldehydes). Nevertheless, it is interesting that the solubility selectivities of heptanal/1-hexene and decanal/1-nonene had no temperature dependence in the case of PDecMS. In the case of OH-PDecMS, however, the selectivity decreased with increasing temperature.

### 3.2. Composite Membrane Characterisation

The selective layers of OH-PDecMS and PDecMS, which were synthesized in the course of this work, were coated onto a flat sheet fluoroplastic porous support MFFK-1. The choice of the MFFK-1 support was determined by its high permeability, which was ensured by a pore size of about 360 nm and a total porosity of about 80% [56]. In particular, it was noted [57] that increasing the surface porosity of the nylon 6.6 MF substrate at the same thickness as the selective layer led to a decrease in the substrate resistance to mass transfer. Accordingly, the MFFK-1 microfiltration substrate provided minimal resistance to mass transfer, thereby reducing the effect of concentration effects at the selective layer–porous substrate interface. Microphotographs of composite membranes obtained by touching the porous support to the surface of the polymer solution are shown in Figure 7.

Thus, composite membranes based on OH-PdecMS and PdecMS with selective layer thicknesses of 5 and 15 μm, respectively, were obtained. These membranes were characterized in terms of gas permeability and permeability of the investigated olefins and aldehydes in the mode of vacuum pervaporation.

#### 3.2.1. Gas Permeation Properties of Composite Membranes

The permeating flux of pure gases through OH-PdecMS and PdecMS composite membranes as a function of pressure drop at 25 °C is presented in Figure 8.

The permeate flux for all gases increased linearly with increasing differential pressure. The flux of CO_2_ for both membranes was greater than the flux of O_2_ and N_2_. This was due to the greater solubility of CO_2_ in the material of the selective layer of the studied membranes. It is worth noting that the difference in the thicknesses of the selective layers of the obtained composite membranes was the reason for the significantly different permeating fluxes. Table 5 shows the permeabilities of the studied composite membranes for CO_2_, O_2_ and N_2_, as well as the selectivities for the CO_2_/O_2_ and CO_2_/N_2_ gas pairs.

The obtained gas transport and separation characteristics of composite membranes were close to the properties of the membrane materials presented in Table 1 and Table 2. Some decrease in selectivity of polysiloxane-based membranes in the transition from dense films to composite membranes is associated with the influence of porous substrate and has been repeatedly noted in the work of researchers [60,61,62]. For composite membranes of OH-PdecMS and PdecMS, the selectivity decreased by 10% and 15%, respectively, in comparison with films. Such a difference in the selectivity drop was probably due to the difference in the thicknesses of the selective layers of the composite membranes. The obtained membranes had sufficiently high permeabilities for carbon dioxide, which demonstrated their potential for use as a preliminary concentrator of CO_2_ from gas mixtures. It should be noted that the obtained selective layer thickness was slightly large for membrane gas separation problems. However, for operation in a membrane contactor, hydroformylation may be of interest in terms of minimizing the transfer of synthesis gas through the membrane.

#### 3.2.2. Mass Transport of Olefines and Aldehydes through Composite Membranes in Vacuum Pervaporation Mode

The transfers of olefins (C_6_-C_9_) and aldehydes (heptanal and decanal) through composite membranes OH-PdecMS and PdecMS were studied in the mode of vacuum pervaporation at temperatures of 30–60 °C from binary solutions with ethanol (20 and 40% wt). The difference in the initial concentrations of aldehydes and olefins was chosen on the basis that during the reaction, olefins are converted into aldehydes and, accordingly, their amount in the reaction mixture should increase. Penetrant fluxes and separation factors are presented in Table 6.

The total flux of binary olefin solutions and the partial flux of olefins was significantly higher for the composite PdecMS membrane. This tendency correlated well with the difference in the thicknesses of the selective layers of these membranes. In the case of binary aldehyde solutions, this trend was not observed. In the case of heptanal, the difference between the values of total and partial fluxes through the OH-PdecMS and PdecMS composite membranes was within 5%. The total and partial decanal fluxes were higher for the OH-PdecMS membrane by ~10% and ~50%. This pattern indicated a strong change in the interaction between aldehyde and the membrane material when -OH groups were introduced into it. The separation factors for olefin/ethanol binary mixtures for the PdecMS membrane showed an enrichment of the permeate in the olefins. Moreover, the increase in the separation factor with a decrease in the molecular weight of the olefin (from 7.2 to 13.8 in the transition from 1-nonene to 1-hexene) was associated with a change in the volatility of these compounds (Table 1). For the OH-PdecMS composite membrane, the separation factor was significantly lower than for the PdecMS composite membrane. Significant enrichment of the permeate with olefins was not observed (the separation factor of C_6_-C_9_ olefins is within 1.3–1.7). This fact was related to the fact that ethanol is not only more volatile and its diffusion in the membrane material is preferable in comparison with the studied olefins, but is characterized by the contribution of hydrogen bonds in the total solubility parameter [55]. For olefins, this contribution is zero. The separation factor for binary aldehyde/ethanol mixtures was less than 1 for both composite membranes. Since C_7_ and C_10_ aldehydes are heavy-bodied compounds with a sufficiently high molecular weight, their diffusion transport through the membrane selective layer was small compared with ethanol. Additionally, low partial vapor pressure of aldehydes at 30 °C did not allow the creation of a high driving force in the process of pervaporation, which also negatively affected the values of separation factors. To exclude the effect of the driving force of the process on the mass transfer through the membrane, permeability values were used. The exclusion of the influence of the difference of the partial pressures of vapors and concentrations allowed the permeabilities of olefins and aldehydes to pass to temperature dependencies, and the comparison of the efficiency of their transfer through the received composite membranes (Figure 9).

Since the permeability of the investigated olefins and aldehydes through the membranes based on polydecylmethylsiloxane is determined mainly by the sorption component, the permeability value decreases with increasing temperature. This trend agreed well with the solubility coefficients of olefins and aldehydes measured in this work in OH-PDecMS and PDecMS (Figure 4). There is also an increase in permeability with increasing molecular weight in the homologous series of penetrants, which also agreed well with the trend in the permeability coefficients of the components. The OH-PDecMS composite membrane was characterized by significantly lower olefin permeabilities compared with the PDecMS composite membrane: for example, the permeabilities at 30 °C and 60 °C for 1-hexene were 43 and 10 mol·m^−2^·h^−1^·bar^−1^ for OH-PDecMS, and 340 and 137 mol·m^−2^·h^−1^·bar^−1^ for PDecMS, whereas for aldehydes, the picture was different: for heptanal, the permeabilities at 30 °C and 60 °C were 2080 and 350 mol·m^−2^·h^−1^·bar^−1^ for OH-PDecMS, and 2025 and 1270 mol·m^−2^·h^−1^·bar^−1^ for PDecMS. The high sorption affinity of OH-PDecMS to aldehydes determined a higher permeability value at 30 °C, but led to considerable plasticization of the material of the selective layer of the composite membrane in the process of pervaporation. Apparently, the plasticization of OH-PDecMS in aldehydes increased with increasing temperature, which led to a lower permeability value at 60 °C and higher, compared with PDecMS. It is worth noting that while in the heptanal/1-hexene pair, both membranes showed selective permeability to heptanal, in the case of the decanal/1-nonene pair, only the OH-PDecMS-based membrane showed selectivity to decanal (Table 7). It was also interesting that the change in aldehyde/olefin selectivity with increasing temperature was different for OH-PDecMS and PDecMS composite membranes. Thus, for OH-PDecMS, the tendency of selectivity decreasing with increasing temperature was observed, which indicated the predominance of the contribution of the sorption component in the selective transport, while for PDecMS, the selectivity increased with increasing temperature, which may be associated with a decrease in diffusion difficulties in the membrane material. Nevertheless, the selectivity values for the OH-PDecMS composite membrane were multiple times higher than for the PDecMS composite membrane, which deems it more attractive for use in hydroformylation membrane reactors.

The apparent permeability activation energies for PDecMS and OH-PDecMS were calculated based on the obtained temperature dependencies of permeability (Table 8).

The negative values of the apparent activation energy demonstrated that the solubility contribution of aldehydes and olefins in the polymer increased with decreasing temperature. This pattern is characteristic of the interaction between polysiloxane membranes and easily condensing penetrant [63,64]. As the molecular weight of olefins and aldehydes in their homologous series increases, the apparent activation energy of their permeability decreases. The apparent activation energies obtained determined the difference in the temperature trends of selectivity presented in Table 7. The permeation activation energy of heptanal (−47.6 kJ/mol) was lower than the activation energy of 1-hexene (−29.8 kJ/mol) for the OH-PDecMS composite membrane, whereas it was higher for the PDecMS composite membrane than for heptanal (−10.3 kJ/mol) and 1-hexene (−25.2 kJ/mol).

## 4. Conclusions

In this work, polydecylmethylsiloxane containing OH-substituted aliphatic groups in the side chain (~7% of the total number of substituents) was synthesized for the first time and the effect of such modification on the transport of acid gases and aldehydes through membranes was demonstrated. It was shown that the CO_2_ gas permeability coefficient of OH-PDecMS was almost two times higher than for the original PDecMS, while the ideal CO_2_/N_2_ and CO_2_/CH_4_ selectivity increased by 6% and 8%, respectively, with modification. The sorption interaction of OH-PDeCMC and PDeCMC with olefins (1-hexene, 1-heptene, 1-octene, 1-nonene) and aldehydes (heptanal and decanal) was studied. The temperature dependence of the permeability coefficients of the liquid components was presented: the solubility coefficient increased with increasing temperature and the molecular weight of the sorbate. The solubility of aldehydes in OH-PDecMS was higher than the solubility of olefins, which determined the high sorption selectivity of this material and the high potential application of this material in membrane hydroformylation contactors. On the basis of the developed material, the composite membrane on the microfiltration substrate MFFK-1 with a thickness of the selective layer of 15 µm was obtained. The thickness of the selective layer of the composite membrane PDetsMS was 5 µm. For composite membranes, transport and separation properties were studied for individual gases (CO_2_, N_2_, O_2_) and liquids (1-hexene, 1-heptene, 1-octene, 1-nonene, heptanal and decanal). The permeability of the obtained membranes was proportional to the thickness of the selective layer. The modified membrane retained an increased selectivity for CO_2_. The predominant transport of aldehydes through the membrane was demonstrated in the study of the transport of aldehydes and olefins in the vacuum pervaporation mode. It was noted that the membrane modified with OH-groups had lower apparent activation energies of aldehyde transport in relation to olefins as compared with the unmodified PDecMS. The negative values of the apparent permeation activation energies were determined by the predominant contribution of the sorption component of transport. The difference in the apparent permeation activation energies for PDecMS and OH-PDecMS membranes determined the change in selectivity with respect to the aldehyde C_n_/olefin C_n-1_ pair.

## Figures and Tables

**Figure 1 polymers-15-00723-f001:**
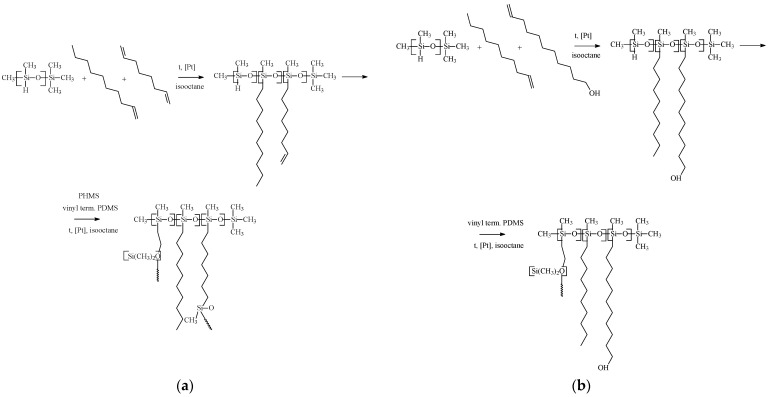
Scheme of hydrosilylation reaction with cross-linked polysiloxanes obtaining: (**a**) PDecMS and (**b**) OH-PDecMS.

**Figure 2 polymers-15-00723-f002:**
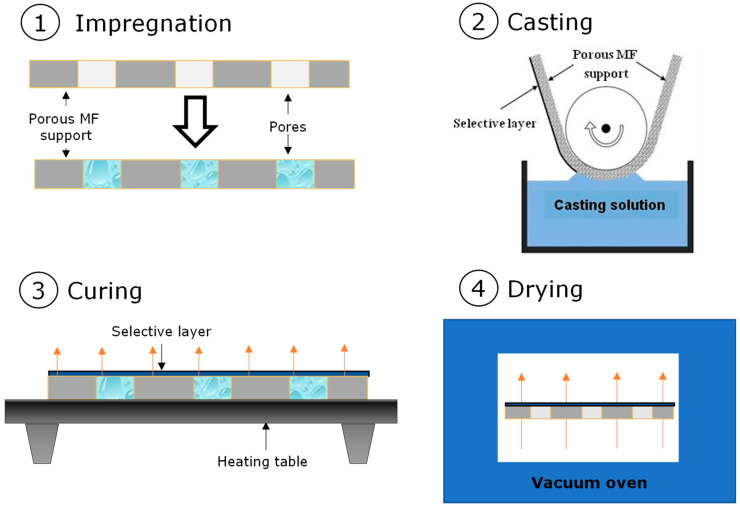
Schematic drawing of composite membrane formation process.

**Figure 3 polymers-15-00723-f003:**
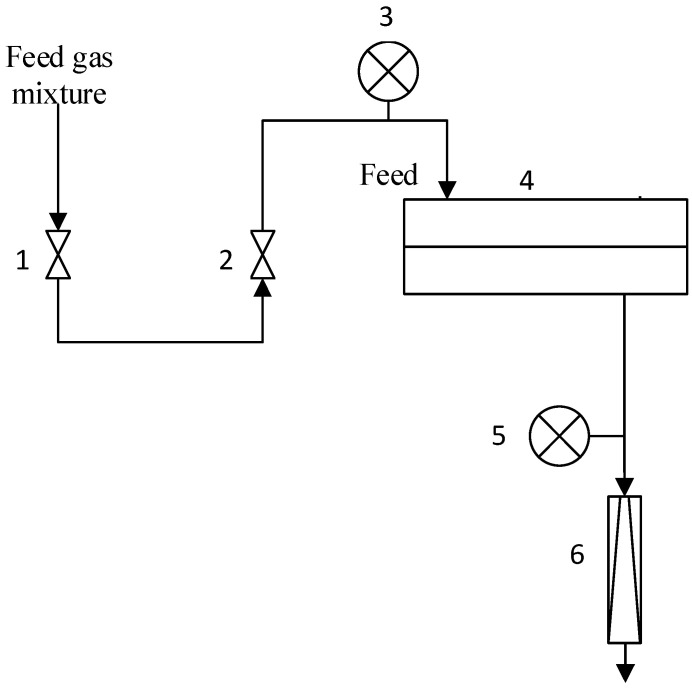
Lab set-up of mix gas permeability measurements: 1—feed on/off valve, 2—pressure regulator, 3—feed pressure gauge, 4—stainless steel module, 5—permeate pressure gauge, 6—permeate bubble mass flowmeter.

**Figure 4 polymers-15-00723-f004:**
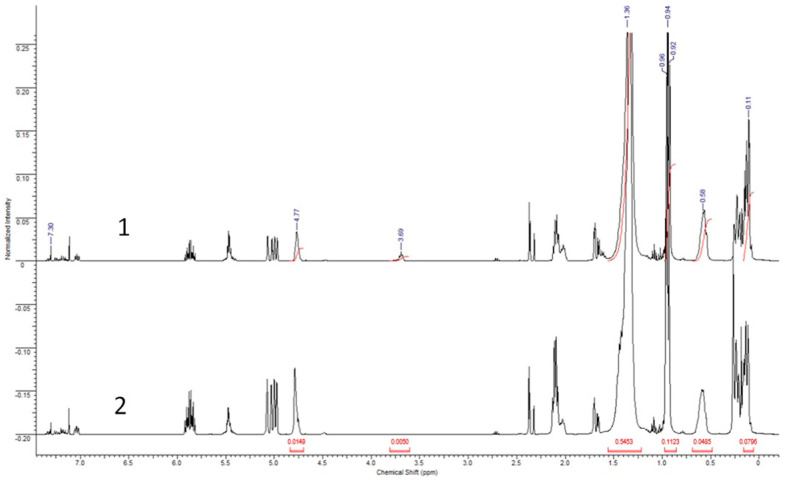
^1^H NMR spectra of OH-PdecMS (1) and PdecMS (2).

**Figure 5 polymers-15-00723-f005:**
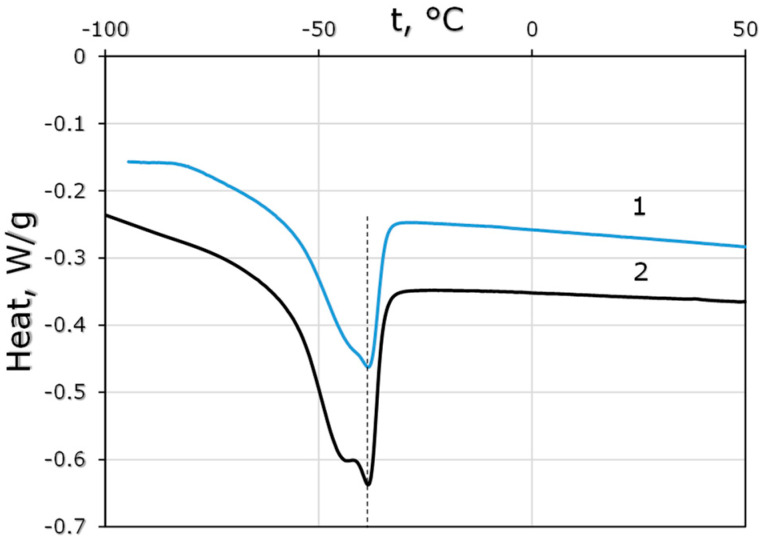
Thermograms of OH-PdecMS (1) and PdecMS (2).

**Figure 6 polymers-15-00723-f006:**
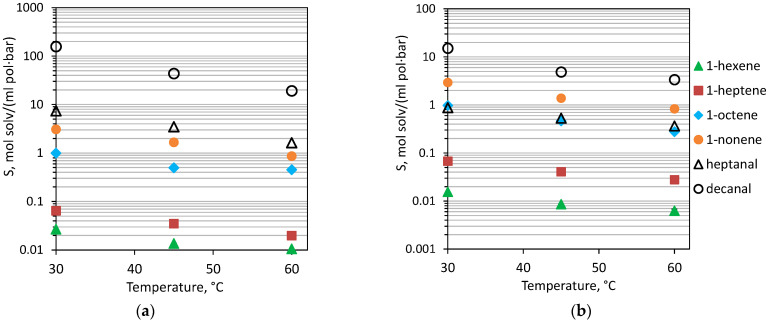
Solubility coefficient (S) of olefines and aldehydes in OH-PDecMS (**a**), and PDecMS (**b**), at 30–60 °C.

**Figure 7 polymers-15-00723-f007:**
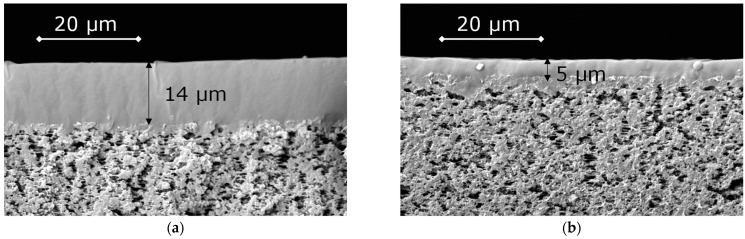
SEM images of OH-PDecMS (**a**), and PDecMS (**b**), composite membranes.

**Figure 8 polymers-15-00723-f008:**
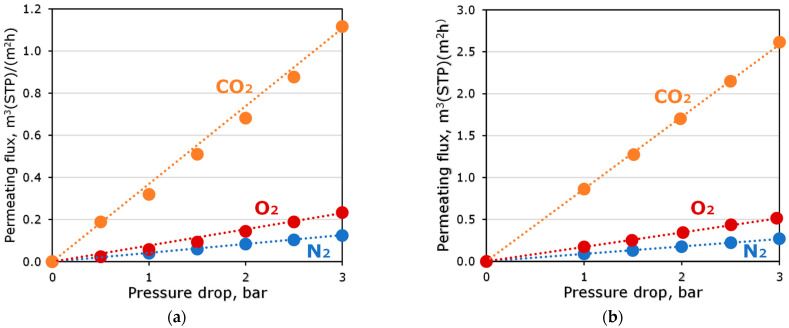
Permeating flux of pure gases through (**a**) OH-PDecMS and (**b**) PDecMS composite membranes as a function of pressure drop at 25 °C.

**Figure 9 polymers-15-00723-f009:**
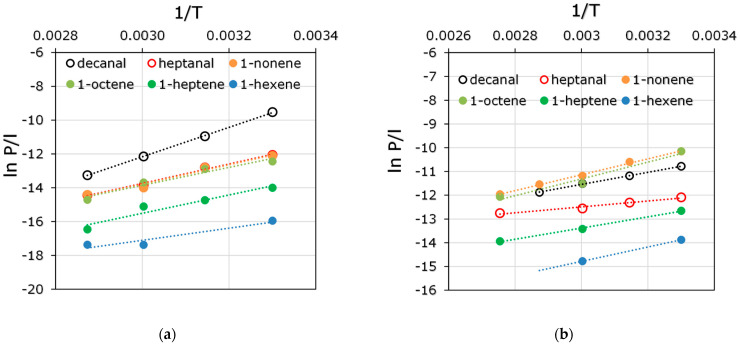
Permeability temperature dependencies of olefines and aldehydes through (**a**) OH-PDecMS, and (**b**) PDecMS.

**Table 1 polymers-15-00723-t001:** Some properties of studied olefines and aldehydes.

Liquid	Molecular Mass, g/mol	Boiling Temperature, °C	Saturated Vapor Pressure at 30 °C, kPa	Saturated Vapor Pressure at 100 °C, kPa
Olefine				
1-hexene	84	63	29.7	290
1-heptene	98	94	9.54	122
1-octene	112	121	3.0	53.7
1-nonene	126	151	1.01	24
Aldehydes				
heptanal	114	153	0.817	19.8
decanal	156	208	0.027	2.21

**Table 2 polymers-15-00723-t002:** Permeability (P), diffusion (D) and solubility coefficients for pure CO_2_, N_2_, and CH_4_, in membrane materials PdecMS and OH-PdecMS (t = 30 °C, *p* = 0.8 bar).

Membrane Material	P, Barrer			D·10^8^, cm^2^/s			S·10^2^, cm^3^/(cm^3^·cm Hg)		
	N_2_	CH_4_	CO_2_	N_2_	CH_4_	CO_2_	N_2_	CH_4_	CO_2_
PdecMS	120	400	1300	1010	420	500	0.12	0.95	2.60
OH-PdecMS	230	770	2700	1050	420	570	0.22	1.83	4.74
PDMS [50]	400	1200	3800	3400	2200	2200	0.12	0.55	1.73

**Table 3 polymers-15-00723-t003:** Ideal permselectivity (α_P_), diffusion (α_D_), and solubility (α_S_) selectivities for CO_2_/N_2_ and CO_2_/CH_4_ gas pairs.

Membrane Material	α_P_		α_D_		α_S_	
	CO_2_/N_2_	CO_2_/CH_4_	CO_2_/N_2_	CO_2_/CH_4_	CO_2_/N_2_	CO_2_/CH_4_
PdecMS	10.8	3.3	0.50	1.19	21.9	2.7
OH-PdecMS	11.7	3.5	0.54	1.36	21.6	2.6
PDMS [50]	9.5	3.2	0.65	1.00	14.7	3.2

**Table 4 polymers-15-00723-t004:** Solubility selectivity of PDecMS and OH-PDecMS at different temperatures.

Liquid Pair	Polymer	Temperature, °C		
		30	45	60
heptanal/1-hexene	OH-PDecMS	440	250	150
	PDecMS	56	60	58
decanal/1-nonene	OH-PDecMS	50	26	22
	PDecMS	4.4	3.5	3.5

**Table 5 polymers-15-00723-t005:** Permeability and permselectivity of PDecMS and OH-PDecMS composite membranes.

Polymer	Permeability, GPU			Permselectivity	
	CO_2_	O_2_	N_2_	CO_2_/O_2_	CO_2_/N_2_
OH-PDecMS	160	30	15	5.3	10.6
PDecMS	320	65	35	4.9	9.2
PDecMS (wet gas) [43]	116	30	15	3.9	7.6
PDMS-ZSM-5 [58]	18	-	-	-	21
PDMS/PAN [59]	3700	860	370	4.3	10

**Table 6 polymers-15-00723-t006:** Permeating fluxes of olefines and aldehydes and separation factors of PDecMS and OH-PDecMS composite membranes at 30 °C.

	Penetrant	OH-PDecMS	PDecMS
Total flux, kg·m^−2^·h^−1^	1-hexene	2.16	7.53
	1-heptene	1.59	6.19
	1-octene	1.36	5.76
	1-nonone	1.02	3.02
	Heptanal	1.42	1.48
	Decanal	0.62	0.49
Component flux	1-hexene	0.75	6.28
	1-heptene	0.68	5.61
	1-octene	0.32	4.44
	1-nonone	0.21	1.23
	Heptanal	0.45	0.42
	Decanal	0.07	0.04
Separation factor (penetrant/ethanol)	1-hexene	1.7	13.8
	1-heptene	1.7	13.4
	1-octene	1.6	10.0
	1-nonone	1.3	7.2
	Heptanal	0.5	0.5
	Decanal	0.2	0.1

**Table 7 polymers-15-00723-t007:** Selectivities of heptanal/1-hexene and decanal/1-nonene for PDecMS and OH-PDecMS composite membranes at different temperatures.

Selectivity	Temperature, °C	OH-PDecMS	PDecMS
heptanal/1-hexene	30	55.9	5.8
	45	39.3	7.7
	60	29.0	9.9
decanal/1-nonene	30	12.3	0.5
	45	7.7	0.6
	60	5.1	0.7

**Table 8 polymers-15-00723-t008:** Apparent permeation activation energy for PDecMS and OH-PDecMS composite membranes.

	Penetrant	OH-PDecMS	PDecMS
Apparent permeation activation energy, kJ/mol	1-hexene	−29.8	−25.2
	1-heptene	−44.8	−19.6
	1-octene	−44.0	−29.5
	1-nonone	−47.5	−27.7
	Heptanal	−47.6	−10.3
	Decanal	−72.3	−21.1

## Data Availability

Not applicable.

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
