# Peer review of "Effect of OH-Group Introduction on Gas and Liquid Separation Properties of Polydecylmethylsiloxane"

_polymers, 2023, doi:10.3390/polym15030723_

Round 1

Reviewer 1 Report

This manuscript discusses the impact of -OH groups on the separation performance of PDecMS membranes. It is overall a good study that can be published in Polymers, however it would benefit from some additional discussion and details prior to publication. I have the following comments/questions:

1) Authors claimed that OH-PDecMS should contain about 7% of OH-groups, do they try other ratios of OH group? How does the ratio of OH groups on the gas separation performance?

2) It would be much appreciated if the authors perform mixed-gas separation experiments to identify the practical applications of the OH-PDecMS membrane?

3) The composite membranes were still rather thick, can the authors fabricate thinner membranes, such as hundred nanometers? 

Author Response

Dear Reviewer, Thank you very much for your time and effort to review our manuscript. We have made our best to improve our manuscript according to your comments. We have expanded Methods, Results and Discussions sections.

1) Authors claimed that OH-PDecMS should contain about 7% of OH-groups, do they try other ratios of OH group? How does the ratio of OH groups on the gas separation performance?

Thank you for your comment. In this work, membranes based on polysiloxanes with OH-containing alkyl side substituents were obtained for the first time. These membranes were obtained with a content of OH groups of about 7%. With the introduction of a larger amount of a substituent containing OH groups, the resulting polymer film becomes heterogeneous, which has a bad effect on its selectivity. This effect is apparently associated with the delamination of the polysiloxane regions substituted with undecanol-1 and decene-1 due to the strong interaction of hydrogen bonds. When choosing the studied concentration of OH groups in the side substituent, the authors proceeded from the fact that an increase in the number of OH groups would increase the selectivity of the membrane with respect to acidic components. A comment about the effect of the number of OH groups on the resulting polymer has been added to the text of the article. «This concentration of undecenol substituents in the polysiloxane makes it possible to obtain a homogeneous polymer film. Increasing the 1-undecenol concentration in the reaction media leads to obtaining heterogeneous material».

2) It would be much appreciated if the authors perform mixed-gas separation experiments to identify the practical applications of the OH-PDecMS membrane?

Thank you for your comment, we will definitely expand this work by measuring the gas mixture in the course of further research.

3) The composite membranes were still rather thick, can the authors fabricate thinner membranes, such as hundred nanometers?

Thank you for your comment. For gas separation membranes, such a membrane thickness of the order of hundreds of nanometers is really attractive; however, for the pervaporation process, a decrease in the thickness of the selective layer can lead to an increase in the concentration polarization and swelling, which leads to a decrease in the separation factor. The aim of our work was to demonstrate the effect of OH groups in the side substituent for various processes (separation of gases and liquids). In the course of further work for these applications, membranes with a thickness optimized for the separation task will be developed. The following comment has been added to the text of the article «It should be note that obtained selective layer thickness is slightly large for membrane gas separation problems. However, for operation in a membrane contactor, hydroformylation may be of interest in terms of minimizing the transfer of synthesis gas through the membrane.»

Reviewer 2 Report

The manuscript "Effect of OH-group introduction on gas and liquid separation properties of polydecylmethylsiloxane" describes the preparation of silicone membranes possessing OH groups and their performance on the selective separation CO2 and aldehydes from gas and olefine mixtures respectively.

The composition of the paper is well designed and sufficient data have been performed. I identified just a few minor issues to be addressed:

1) in Section 2.2 Membrane materials synthesis it should be added the schemes of reaction between the silicone precursors and a graphical representation of the membrane preparation in Section 2.3.

2)The technique of sorption measurements should be briefly presented, not only as reference [9]

3) in section 3.1.1 How the OH content was calculated? The values of integrals are missing in Fig. 2. The NMR signals are not appreciated by their intensity, but through the integrals associated with each signal. The NMR section should be rewritten, there are many errors, and the scientific soundness is missing.

4) Table 3 should be moved in Materials section.

5) In Tables 4, 5, 6, 7 and 8 some comparative data must be added.

6) In section 3.2 it is not clear how the composite membranes have been obtained. How the porosity was created?

I recommend the publication of this paper after minor revision.

Author Response

Dear Reviewer, 

Thank you very much for your time and effort to review our manuscript. We have made our best to improve our manuscript according to your comments. We have expanded Methods, Results and Discussions sections. 

  • in Section 2.2 Membrane materials synthesis it should be added the schemes of reaction between the silicone precursors and a graphical representation of the membrane preparation in Section 2.3.

Thank you for your comment. This illustration were added to article text (Fig. 1 and 2).

2)The technique of sorption measurements should be briefly presented, not only as reference [9]

We added some explanation for sorption measurement according with your comment

“The technique is described in detail in [44] and shortly represented below. Polymer film with fixed mass m0 was exposed in solvent mixture for the 48 hour in isothermal condition.”

Sorption measurements and calculation were described in lines 213-248.

3) in section 3.1.1 How the OH content was calculated? The values of integrals are missing in Fig. 2. The NMR signals are not appreciated by their intensity, but through the integrals associated with each signal. The NMR section should be rewritten, there are many errors, and the scientific soundness is missing.

Thank you for your comment. We rewrite NMR section in Results and add integral values. The OH-group content was calculated based on the material balance of the synthesis and NMR data. From NMR data, to identify ration undecyl-1-ol to decyl in OH-PDecMS the peak integrals were calculated for shifts 0.94 (-CH3 in aliphatic chain) and 3.69 (-CH2- in aliphatic chain bonded with -OH group). Their values were 0.1123 and 0.0050, respectively. It should be taken into account that peak integral heeds to be normalized of H atom amount in considered group (0.1123/3 = 0.0374 and 0.0050/2 = 0.0025). So, there the amount of OH-substituted side groups was 6.3% in OH-PDecMS (0.0025/(0.0025+0.0374)). The value obtained correlates well with the mass balance of the synthesis (6.5 mol. % undecyl-1-ol groups).

4) Table 3 should be moved in Materials section.

According to your comment, we transfer this table to Materials section.

5) In Tables 4, 5, 6, 7 and 8 some comparative data must be added.

Thank you for your comment. In table 5 we added some data for comparison. In case tables 4, 6 and 7 there is no possibility to add comparative data, because they will not be informative. When studying the pervaporation characteristics of membranes, separation conditions, including the composition of the mixture to be separated and the components to be isolated, are of great importance. For the studied aldehydes and olefins, there are no comparable experimental data for polysiloxane membranes in the literature for comparison.

6) In section 3.2 it is not clear how the composite membranes have been obtained. How the porosity was created?

The first sentence of this section has been changed to avoid confusion. In section 3.2 composite membrane was obtained by kiss-coating technic: porous support MFFK-1 touched the surface of the polymer solution to form a meniscus. Thus, a non-porous membrane was obtained. Section 3.2 discusses the porosity of MFFK-1 in the context of its resistance to mass transfer and the rationale for its choice as a substrate.

Round 2

Reviewer 1 Report

The manuscript has been sufficiently improved to warrant publication in Polymers.